# Calcium: A Critical Factor in Pollen Germination and Tube Elongation

**DOI:** 10.3390/ijms20020420

**Published:** 2019-01-19

**Authors:** Ren Hua Zheng, Shun De Su, Hui Xiao, Hui Qiao Tian

**Affiliations:** 1The Key Laboratory of Timber Forest Breeding and Cultivation for Mountainous Areas in Southern China, Fujian Academy of Forestry, Fuzhou 350012, China; ssdforest@163.com (S.D.S.); huix01@163.com (H.X.); 2School of Life Sciences, Xiamen University, Xiamen 361102, China

**Keywords:** calcium, pollen germination, pollen tube growth, stigma, style

## Abstract

Pollen is the male gametophyte of higher plants. Its major function is to deliver sperm cells to the ovule to ensure successful fertilization. During this process, many interactions occur among pollen tubes and pistil cells and tissues, and calcium ion (Ca^2+^) dynamics mediate these interactions among cells to ensure that pollen reaches the embryo sac. Although the precise functions of Ca^2+^ dynamics in the cells are unknown, we can speculate about its roles on the basis of its spatial and temporal characteristics during these interactions. The results of many studies indicate that calcium is a critical element that is strongly related to pollen germination and pollen tube growth.

## 1. Introduction

The fertilization process of high plants begins with pollen grains landing on the stigma and germination of the pollen tubes. If the pollen is compatible, then pollen tubes grow through the pistil tissues of the stigma and style, over the surface of the placenta, and then through the micropyle of the ovule to reach the female gametophyte in the embryo sac. Two male gametes are discharged from the pollen tube into the degenerated synergid where one fuses with the egg cell and the other fuses with the central cell. A series of interactions occur between male and female cells and tissues while pollen tubes are elongating through the diploid sporophytic tissue of the pistil. It is these interactions that ensure the pollen tube enters the embryo sac and that both deposited sperm cells fuse with the egg and central cell successfully. The role of the calcium ion (Ca^2+^) in these interactions has received particular attention, as it appears to provide a universal signal with pleiotropic effects on attraction, long and short-distance communication, cellular fusion, and cell signaling [1,2,3].

Pollen is the male gametophyte of angiosperms and the only cell that can move and leave the parent plant body. Although pollen is very simple in its composition (2 or 3 cells), its germination and pollen tube growth are very complex mechanisms that ensure the tube reaches its target cell. The results of many studies indicate that Ca^2+^ dynamics in pollen and the gynoecium are important for these mechanisms to operate (Figure 1) (discussed in detail later). Calcium ions show different temporal and spatial distribution in the cells of male and female tissues during development. We can speculate on the functions of Ca^2+^ in controlling pollen germination and pollen tube growth on the basis of its temporal and spatial changes in reproductive cells and organs. Although there are continually reports about control mechanisms of pollen tube growth [4,5,6,7], the most results are from in vitro assays. In this review, we focus on the functions of Ca^2+^ in pollen germination and pollen tube growth in the stigma and style tissues of higher plants. We discuss the relationships between tissue-level changes in Ca^2+^ uptake and pollen and style maturation during the progamic phase.

## 2. Characteristics of Pollen Tube Growth

When pollen grains land on the stigma, compatible grains germinate and an elongating tube forms and grows through the aperture of the pollen grain. Pollen germination is closely related to its hydration to produce turgor, which pushes the pollen cytoplasm to protrude through the aperture to form the pollen tube. Pollen tube growth is both polar and directional. The polar growth of the tube is limited to its tip, and is closely related to the tube’s internal structure [8]. The bipartite pollen tube wall consists of an inner callose sheath and an outer region of pectin-rich fibrillar layers. The pollen tube tip comprises a single pectin layer [9]. Immunocytochemical studies have labelled at least two types of pectin in the pollen tube wall: polygalacturonic acid pectin and methyl-esterified pectin. Acid pectins capable of crosslinking Ca^2+^ are localized in the subapical region of the pollen tube, whereas richly methyl-esterified pectins are abundant at the tube tip. Calcium-crosslinked pectin in the subapical region of the pollen tube becomes a rigid and insoluble gel. The tube tip region consists of richly methyl-esterified pectins, and is the most elastic region in the tube. It is likely to be the expansion point under internal turgor pressure [10].

During pollen tube formation, almost the entire contents of the pollen grain move into the tube. Within the growing tube, most of the cytoplasm is confined to the apex, and a large vacuole fills the grain and the older region of the tube. To restrict the cytoplasm to the apical region of the growing tube, a series of callose plugs are formed at a regular distance behind the tip [11]. The formation of vacuoles in the tube cytoplasm is a persistent process that maintains tube turgor during pollen germination and tube growth [12]. The vacuoles move to the basal part to push the cytoplasm to the apex of the tube. Thus, the pollen tube shows polar growth that is mediated by this internal movement mechanism and the composition of the pollen wall.

Pollen tubes grow down the style and then precisely target a single cell within the female embryo sac. Molecular changes that occur before pollen tube curvature include protein interactions between male and female cells [13,14,15], but Ca^2+^ also plays a role in determining the orientation of tube growth [16,17], as discussed in detail later. Pollen germination and pollen tube growth are special processes during the early stage of fertilization, and require unique control mechanisms, including a series of interactions among the pollen, pollen tube, pistil cells, and other tissues. Calcium dynamics in pollen and pistil cells represent an interaction between pollen and pistil cells.

## 3. Calcium Distribution in the Pollen Germination

When pollen grains germinate in small numbers in vitro, the germination and elongation rates are often slower than those observed in larger-scale settings. This population effect can be overcome by adding more pollen, or by adding Ca^2+^ to the medium. This demand for Ca^2+^ for pollen germination was confirmed for 86 species in 39 plant families [18], indicating that Ca^2+^ dependency is widespread among flowering plants. The pollen grains of some plants can germinate on media without Ca^2+^; however, it is presumed that some Ca^2+^ stored in the pollen wall is released into the medium during pollen hydration. In rice, for example, pollen grains contain abundant Ca^2+^ in the cell walls before germination [19].

External Ca^2+^ not only affects pollen germination but also pollen tube growth. Mascarenhas and Machlis (1962) observed a chemotropic response of *Antirrhinum majus* pollen tubes to an external Ca^2+^ gradient in vitro [20]. In another study, lower Ca^2+^ concentrations appeared to decrease tube elongation via excess accumulation of vesicles at the tip, inducing apical swelling. Increased Ca^2+^ concentrations appeared to accelerate vesicle fusion at the pollen tube tip, but may have also altered cytoskeletal elements to contribute to a thickened wall at the tube tip [12]. The results of many early studies indicated that Ca^2+^ is required for the processes of pollen germination and pollen tube growth [1,15].

## 4. Calcium Distributions in Pollen Tubes

Calcium shows special distribution patterns in the pollen tube. In pollen tubes of *Lilium longiflorum*, concentrations of free Ca^2+^ estimated using quin-2 fluorescence were ~10^−7^ mol/L near the tip and 2 × 10^−8^ mol/L at the base [21]. The free Ca^2+^ concentrations measured using the fura-2 fluorescence indicator were 1.7 to 2.6 μM at the tip and 60 to 100 nM at 100 μm behind the tip [22]. The membrane-impermeable injected label fura-2 dextran estimated the Ca^2+^ concentration to be approximately one-third lower at the tube tip (490 nM at the extreme tip) and 170 nM at 10–20 μm behind the tip. The free Ca^2+^ gradient was observed to drop off sharply within 10–20 μm of the tube tip [23]. Although the estimated values of free Ca^2+^ differed markedly among those reports, a free Ca^2+^ gradient restricted to the apical 20 μm of the pollen tube tip was consistently detected, regardless of the technique. Changes in free Ca^2+^ beyond the first 20 μm of the pollen tube were simply not observed [24,25,26,27,28]. In all of those studies, the Ca^2+^ gradient was detected only in growing pollen tubes. Upon death or the addition of Ca^2+^ channel blockers or antagonists to the medium, the Ca^2+^ gradient disappeared from the tip of the pollen tube, and tube growth stopped. When such arrested tubes were transferred to medium without Ca^2+^ blockers or antagonists, they rapidly reestablished a Ca^2+^ gradient and elongation was reinitiated. The strongly concentrated Ca^2+^ gradient in the pollen tube tip corresponded closely with vigorous tip elongation [7].

## 5. Calcium Gradient Oscillation and Pulsatory Growth of the Pollen Tube

Pierson et al. (1994) used more rapid ratiometric calculations based on confocal microscopy data to confirm the steepness and depth of the tip-focused intracellular Ca^2+^ gradient in lily pollen tubes, which ranged from >3.0 μM at the apex to 0.2 μM within 20 μm from the tip [29]. Although the levels of intracellular Ca^2+^ were assumed to remain essentially stable over time, further analyses based on measurements using a Ca^2+^-specific vibrating electrode revealed that extracellular Ca^2+^ entered the pollen tube at influx rates varying between 1.4 and 14 pmol cm^−2^ s^−1^. These rates of extracellular Ca^2+^ influx into the pollen tube were uneven but reasonably periodic. The tubes displayed pulsed growth, with pulses of pollen tube elongation that corresponded with the periodic deposition of cell wall components [30]. Ratiometric ion imaging of the intracellular Ca^2+^ gradient indicated that the high point of the gradient fluctuated in magnitude from 0.75 to > 3μM during the 60-s measurement intervals. The elevation of the Ca^2+^ gradient appeared to be correlated with an increased rate of tube growth [27]. Holdaway-Clarke et al. (1997) also found that the tip-focused Ca^2+^ gradient oscillated with the same periodicity as pollen tube growth, but the pulses were slightly out of phase. The extracellular Ca^2+^ influx was delayed by about 11 s compared with the oscillation of the Ca^2+^ gradient [26].

The relationships among Ca^2+^ oscillations, pulsed tube growth, and extracellular Ca^2+^ influx involve dynamic spatial and temporal characteristics of Ca^2+^ signaling. Pulsed pollen tube elongation, however, is not a requirement for normal pollen tube elongation, even in plants with pollen tubes that show characteristic pulse growth. Geitmann and Cresti (1998) found that the inorganic Ca^2+^ channel inhibitors La^3+^ and Gd^3+^ caused pulsating pollen tubes to abandon their rhythm and elongate steadily. The organic inhibitors of Ca^2+^ channels, nifedipine and verapamil, slowed the pulse frequency, but did not inhibit pulsed growth [31]. These results strongly suggested that at least two types of Ca^2+^ channels are present in the plasma membrane of the tube tip. When apical elongation of the pollen tube was inhibited, periodic membrane traffic still occurred with nearly the same periodicity as that during normal tube elongation [32]. Dynamic Ca^2+^ concentrations play a central role in increasingly complicated models of growth oscillations in pollen tubes, both in a direct physiological role and more importantly, in a signaling role [6,7,33]. The occurrence of Ca^2+^ oscillations in pollen tubes has become an important model for understanding how such oscillations regulate growth in higher plants.

## 6. Calcium and Turgor Formation during Pollen Tube Growth

The production and maintenance of internal turgor in pollen and the pollen tube provide the driving force for pollen germination and tube growth. However, it is unknown how turgor formation is controlled when pollen absorbs water (hydration). During pollen germination, turgor is produced by the formation of vacuoles. Although some reports have described vacuole formation during microspore development [34,35,36,37], less is known about vacuole metabolism during pollen germination. The rate and type of vacuole metabolism differ among developmental stages (e.g., between the microspore and mature pollen). Vacuole metabolism is a developmental continuation in the microspore, but a new development in mature pollen.

Similar to pollen germination, the pollen tube also requires turgor to push the cytoplasm to the front of the tube. In a growing tube, most of the cytoplasm is confined to the apex region, and large vacuoles fill the older region. Vacuole metabolism produces turgor in the pollen tube. In vigorously growing pollen tubes, vacuole formation is active and continuous [12]. However, it is unknown how vacuoles are generated in pollen and the pollen tube. Recent studies on wheat [37] and *Bauhinia blakeana* [38] showed that Ca^2+^ affects the composition of the microspore cytoplasm, primarily by accumulating in mitochondria and destroying their inner membranes (cisterns) to form small vacuoles. These vacuoles then expand and fuse to become large vacuoles during microspore vacuolisation. However, more research is needed to explore the similarities and differences in vacuole formation among plant species.

## 7. Calcium Dynamics and Reorientation of Pollen Tube during Elongation

The ability of the pollen tube to reorient tip growth is an interesting phenomenon. Pollen tubes precisely target a single cell within the female embryo sac, and the molecular changes that occur before pollen tube curvature are a topic of great interest. Malhó’s group showed that the directionality of *Agapanthus umbellatus* pollen tubes could be modified by iontophoretic introduction of Ca^2+^ and by weak electrical fields, which caused pollen tubes to elongate toward the cathode. Introducing a localized gradient of the ionophore A23187, which is believed to open Ca^2+^ channels, caused the pollen tube tip to reorient towards A23187. When the Ca^2+^ channel blocker GdCl_3_ was added to the growth medium, the pollen tube tips elongated away from the GdCl_3_. An accumulation of cytosolic free Ca^2+^ preceded the reorientation of the tip and predicted the location of future elongation [39]. A further demonstration of the effect of free Ca^2+^ was obtained by microinjecting caged Ca^2+^ into living pollen tubes. The injected pollen tubes were irradiated eccentrically with ultraviolet light near their tip, causing photolysis of the cages and the release of free Ca^2+^ at that location. The resulting transient rise in free Ca^2+^ induced a reorientation of tip growth towards the irradiated site. Tip growth resumed near the irradiated region and caused a sustained reorientation of the elongating tip. The site of tip reorientation corresponded closely with the local release of Ca^2+^. This pattern was reinforced by a decline in Ca^2+^ levels on the opposite side of the tube, completing the reorientation [27,40]. Thus, Ca^2+^-rich areas within a gradient can reorient tip elongation, thereby establishing the directionality of future pollen tube elongation. Related studies demonstrated that a kinase present in the pollen tube apex might also be involved in regulating localized Ca^2+^ channel activity [41,42].

## 8. Calcium Distribution in the Stigma

The stigma and style of higher plants are the pathway of the growing pollen tube. The characteristics of Ca^2+^ distribution in the stigma and style reflect the interaction between pistil tissue and the regulation of pollen germination and tube growth.

The stigmas of higher plants intercept pollen grains from many sources, and are the first site where pollen screening occurs. Stigmas are diverse and vary widely among plant species. When pollen grains land on the stigma, compatible grains germinate and pollen tubes elongate from them. The process of pollen germination is related to Ca^2+^ metabolism of pistil tissues. Tirlapur and Shiggaon (1988) found abundant membrane Ca^2+^ in the papillae of *Ipomoea batatas* using chlorotetracycline (CTC) [43]. Bednarska (1989) confirmed this result in *Ruscus aculeatus* [44] using CTC and X-ray microanalysis, and further observed that germinating pollen of *Primula officinalis* and *R. aculeatus* absorbed Ca^2+^ from the stigma [45]. Studies using antimonate precipitation indicated that, in sunflower, Ca^2+^ was more abundant on the receptive surfaces of the stigma, especially outside and inside the papillae, than on non-receptive surfaces [46]. Abundant calcium precipitates were also detected in the intercellular matrix of stigmatic tissues of cotton [47] and on the surface of *Brassica oleracea* stigma after pollination, particularly where pollen grains had landed and tubes had germinated on the stigmatic papillae [48]. Ge et al. (2009) analyzed pollen tube growth in the tobacco stigma. In tobacco, the stigma is a rod-like structure with an enlarged top, and it becomes wet with a thick layer of glycoprotein exudate at anthesis. This layer contains abundant vesicles [49]. Ge et al. (2009) detected abundant calcium precipitates in these vesicles. When pollen grains arrived at the tobacco stigma, they became hydrated and swelled. The accumulation of Ca^2+^ precipitates at the pollen aperture suggested that the germinating pollen grains absorbed Ca^2+^. The calcium precipitates that accumulated in pollen cytoplasm were initially concentrated within small vacuoles. The small vacuoles fused as germination proceeded to form prominent large vacuoles, which created the turgor to push the cytoplasm along the forming pollen tube [49]. The lettuce stigma is composed of two, dry slivers, each with a receptive plane consisting of papillae cells, and a non-receptive plane. Before anthesis, few calcium precipitates were visible in the papillae cells. At anthesis, abundant calcium precipitates accumulated in the wall between papillae cells, the site of pollen tube penetration into the stigma. In the epidermal cells of the non-receptive portion of the stigma, few calcium precipitates were observed in the cell and cell walls [50]. These observations indicated that absorption of Ca^2+^ by pollen grains is related to vacuole genesis. The accumulation of abundant Ca^2+^ in and on stigmatic surfaces was consistent with the need for extracellular Ca^2+^ for pollen germination and tube elongation in vitro, providing further evidence that the pistil affects pollen growth in vivo.

## 9. Calcium Distribution in the Style

After pollen germinates, the tube penetrates the stigmatic surface and grows into the transmitting tissue of the style. Mascarenhas and Machlis (1962) found a chemotropic response of pollen tubes to Ca^2+^ and observed a gradient of Ca^2+^ concentrations from the stigma to the ovule in the pistil of *A. majus*. On the basis of in vitro assays, they speculated that the pollen tube responds to a Ca^2+^ gradient in the pistil, which attracts the tube to elongate towards the ovary [20]. However, Ca^2+^ gradients were not detected in several other examined flowering plants [51,52]. After those reports, few studies focused on this topic. However, the development of a more sensitive technique to detect Ca^2+^ in multicellular tissues (i.e., precipitation of Ca^2+^ using antimonate) led to renewed interest in Ca^2+^ distribution in the stigma and style. In *Brassica napus* [53], *Helianthus annuus* L. [46], and *Gossypium hirsutum* L. [47], abundant Ca^2+^ stores were detected in the transmitting tissues. In these plants, calcium-induced antimonate precipitates were mainly detected in the apoplastic system of the transmitting tissue, i.e., the intercellular matrix and cell wall, where pollen tubes elongate. Calcium is an essential requirement for pollen tube growth in vitro. Similarly, the abundant Ca^2+^ in the transmitting tissue of the style was suggested to meet the requirements for pollen tube growth in vivo. In tobacco, abundant calcium precipitates were detected in the stigma at anthesis, but less were detected in the transmitting tissue from anthesis until 11 h after pollination. At 22 h after pollination, Ca^2+^ concentrations were found to increase distally from the stigmatic interface with the transmitting tissue through the length of the style to the ovary, resulting in the formation of a Ca^2+^ gradient in the transmitting tissue [49]. In lettuce, Ca^2+^ formed a gradient from the top to the base of the style in the transmitting tissue and parenchyma cells before pollination. After pollination, the Ca^2+^ levels increased in the transmitting tissue, and the gradient distribution became stronger. A Ca^2+^ gradient was also detected in the tracheae of the vascular bundle of the style. These results indicated that pollination induces an increase in stylar Ca^2+^ levels [50].

## 10. Calcium Dynamic in Self-Incompatibility Response

Calcium not only promotes, but also inhibits pollen tube growth. In self-incompatible plants, Ca^2+^ dynamics are related to the self-incompatibility response. When S-glycoprotein was extracted from self-incompatible stigmas of *Papaver rhoeas* and added to the culture medium, the Ca^2+^ concentration increased in pollen tubes and their elongation was inhibited. However, when either compatible or heat-denatured incompatible stigmatic S-glycoprotein was introduced into the culture medium, the Ca^2+^ levels did not increase in pollen tubes and elongation continued normally [54]. When caged Ca^2+^ was introduced into pollen tubes, their elongation also proceeded normally. However, cage photolysis resulted in a similar inhibition of tube elongation, which was related to the artificially elevated internal Ca^2+^ levels. Photoactivation of caged InsP_3_ elicited a similar response [55]. To eliminate the effect of potential contamination by other stylar components, the SI-gene was cloned and expressed in *Escherichia coli*. The purified gene product elicited the same result. Thus, the S-protein alone was found to be capable of triggering the Ca^2+^ signal during the pollen self-incompatible response [56].

The direct imaging of Ca^2+^ confirmed that the addition of S-protein resulted in an influx of extracellular Ca^2+^ at the “shank” of the pollen tube. The influx of extracellular Ca^2+^ was confirmed to play a role in the self-incompatible response [57], because the addition of the Ca^2+^-antagonists Verapamil or the Ca^2+^ channel blocker La^3+^ allowed pollen to overcome the self-incompatible response and elongate into the style [58]. The Ca^2+^ influx was found to differ between compatible and self-incompatible plants, further confirming the importance of extracellular Ca^2+^ influx in the self-incompatible response [59]. In *Nicotiana alata*, self-pollen tube elongation was inhibited by a ribonuclease in the stylar transmitting tissue [60]. Presumably, Ca^2+^-dependent protein kinases from the pollen tube activated the S-ribonuclease in the transmitting tissue, resulting in the digestion of RNA in the pollen tube and the inhibition of tube growth [61]. Immunological studies have suggested that calmodulin, calmodulin-like, and calreticulin-like proteins are involved in Ca^2+^ related cell signaling during pollen–pistil interactions [62,63].

Recent studies on the self-incompatibility response of higher plants have revealed interesting characteristics of Ca^2+^ distribution in stigma tissues. Iwano et al. (2014) found that, in *Brassica rapa*, the compatible pollen (cross) induced a greater Ca^2+^ increase in papilla cells of the stigma, while incompatible pollen (self) induced a smaller Ca^2+^ increase. They speculated that the compatible pollen coat contains signaling molecules that stimulate Ca^2+^ export from papilla cells [64]. Iwano et al. (2015) used self-incompatible *Arabidopsis thaliana* expressing S-locus (Self-incompatibility) protein 11 (SP11), S-locus cysteine-rich protein (SCR), and S-receptor kinase, and found that self-pollination specifically induced an increase in cytoplasmic Ca^2+^ in papilla cells. Direct application of SP11/SCR to the papilla cell protoplasts induced a Ca^2+^ increase, which was inhibited by the glutamate receptor channel blocker AP-5. An artificial increase in Ca^2+^ in papilla cells arrested hydration of wild-type pollen. Treatment of papilla cells with AP-5 interfered with self-incompatibility, and the Ca^2+^ increase during the self-incompatibility response was reduced in gene mutants of the glutamate receptor-like channel. It was speculated that the Ca^2+^ influx mediated by the glutamate receptor-like channel is essential for the self-incompatibility response leading to self-pollen rejection [65]. The results of those two studies revealed different Ca^2+^ dynamics; that is, an increase in the Ca^2+^ concentration in papilla cells during the self-incompatibility response in *A. thaliana*, but not in *B. rapa.* These differences indicate that the self-incompatibility response in higher plants is complex and diverse. Also, both *B. rapa* and *A. thaliana* have papilla cells in their stigmas. Further studies are required to explore Ca^2+^ dynamics in stigmas without papilla cells.

## 11. Calcium Dynamics during Interaction of Pollen Tube with Synergids

Synergid cells in the embryo sac are the final destination for pollen tube growth. The attraction function of the tube entrance of synergids is a hot point in plant reproductive biology. Early studies investigated Ca^2+^ dynamics in ovaries and ovules of wheat [66], pearl millet [67], tobacco [68], *B. napus* [69], *Plumbago zeylanica* [70], and rice [71]. These studies revealed high levels of Ca^2+^ in the synergids, which may function to attract the pollen tube to the embryo sac. Using *Arabidopsis* expressing the GFP-based Ca^2+^-sensor yellow chameleon 3.60 (YC3.60) in pollen tubes and synergid cells, Iwano et al. (2012) investigated Ca^2+^ dynamics in ovules during pollen tube guidance and reception. In the pollen tube growing towards the micropyle, pollen tubes started turning within 150 µm of the micropylar opening; the Ca^2+^ concentration was higher in these pollen tube tips than in those not growing towards an ovule. The results suggested that attractants secreted from the ovules affect Ca^2+^ dynamics in the pollen tube [72].

As the end point of pollen tube growth, the synergids are the site where pollen tube growth must stop. Still in *A. thaliana*, the Ca^2+^ concentration in synergid cells did not change when the pollen tube grew towards the micropyle or entered the ovule. When the pollen tube arrived at the synergid cell, a Ca^2+^ oscillation was produced at the micropylar pole of the synergid, and spread towards the chalazal pole. Finally, the Ca^2+^ concentration in the synergid cell peaked at pollen tube rupture. The authors speculated that signals from the pollen tube induced the Ca^2+^ oscillations in synergids, and the peak Ca^2+^ content in synergids induced pollen tube rupture, indicating an interaction between the pollen tube and the synergids [72]. This was the first experimental evidence that synergids can arrest pollen tube growth by calcium dynamic. Then, Hamamura et al. (2014) found that after pollen tube discharge and plasmogamy, the egg and central cells of *A. thaliana* displayed transient spikes in Ca^2+^ concentrations, but not oscillations. In contrast, the synergid cells displayed Ca^2+^ oscillations on pollen tube arrival. The two synergid cells showed distinct Ca^2+^ dynamics depending on their respective roles in tube reception [73]. Generally, there are two synergids beside the egg in the embryo sac: the degenerated synergid that accepts the pollen tube, and the persistent synergid. An experiment using two genetically encoded Ca^2+^ sensors with non-overlapping emission spectra showed that the Ca^2+^ oscillations increased in only one synergid following pollen tube entrance [74]. The synergids controlled this process by coordinating their distinct Ca^2+^ signatures in response to the Ca^2+^ dynamics and the growth behavior of the pollen tube. In addition, the Ca^2+^ signatures were interchangeable between the two synergids, implying that their fates of death or survival were determined by reversible interactions with the pollen tube [74]. Denninger et al. (2014) reported that Ca^2+^ oscillations were initiated in synergid cells after physical contact with the pollen tube apex. In egg and central cells, a short Ca^2+^ transient was associated with pollen tube burst and sperm cell arrival. A second extended Ca^2+^ transient solely in the egg cell was correlated with successful fertilization [75]. In all of those studies, interactions between the pollen tube and synergids were found to be mediated by Ca^2+^ dynamics in both cells, confirming that the synergids function in attracting pollen tube entrance and stopping pollen tube growth [76,77].

## 12. Conclusions and Prospects

Calcium distribution in the pollen tube is closely related to pollen tube function, polarity, and tip elongation. The asymmetrical Ca^2+^ distribution in the tube tip triggers reorientation of the axis of elongation. The subcellular distribution of Ca^2+^ in the pollen tube occurs along a steep dynamic gradient that is related to tube elongation. The most pistils of flowering plants contain abundant Ca^2+^ that meets the requirements for tube elongation in vivo and attracts pollen tubes to the ovule and embryo sac. Besides the spatial and temporal characteristics of Ca^2+^ distribution in the stigma and style, the Ca^2+^ dynamics in papilla cells of the stigma and pollen also indicate a function in controlling the self-incompatibility response. High levels of Ca^2+^ in degenerated synergids not only guide the entry of the pollen tube, but also stop tube growth. All of these results indicate that Ca^2+^ plays critical roles in pollen tube growth in vivo.

Significant progress has been made in understanding the role of Ca^2+^ in pollen tube growth, but numerous questions remain unanswered. First, the exact function of Ca^2+^ in the growing pollen tube has not been proven. Some reports have indicated that Ca^2+^ associates with the pollen tube wall to confer rigidity and mechanical strength, but does not drive pollen tube growth. The origin of tube growth is its internal turgor, which is related to vacuole genesis. However, the genesis of vacuoles in the pollen tube has not been confirmed by any assay method. Second, pollen tubes grow in plant pistils in vivo, but many reported results were obtained in vitro. In vitro conditions are assumed to mimic the natural state of pollen tubes and represent gamete behavior, but significant and perhaps critical functional differences may be overlooked. Pollen tubes elongate faster, tubes live longer, and sperm cells (in bicellular species) form sooner and more reliably in vivo than in vitro. Pollen tube growth in vivo involves interactions with pistil tissues, especially at critical stages such as tube reorientation and the cessation of growth. Breaking of the pollen tube also involves an interaction between the pollen tube and the synergids mediated by Ca^2+^. Confirmation of the roles of Ca^2+^ in pollen tube growth in vivo would reinforce and extend the current state of knowledge.

## Figures and Tables

**Figure 1 ijms-20-00420-f001:**
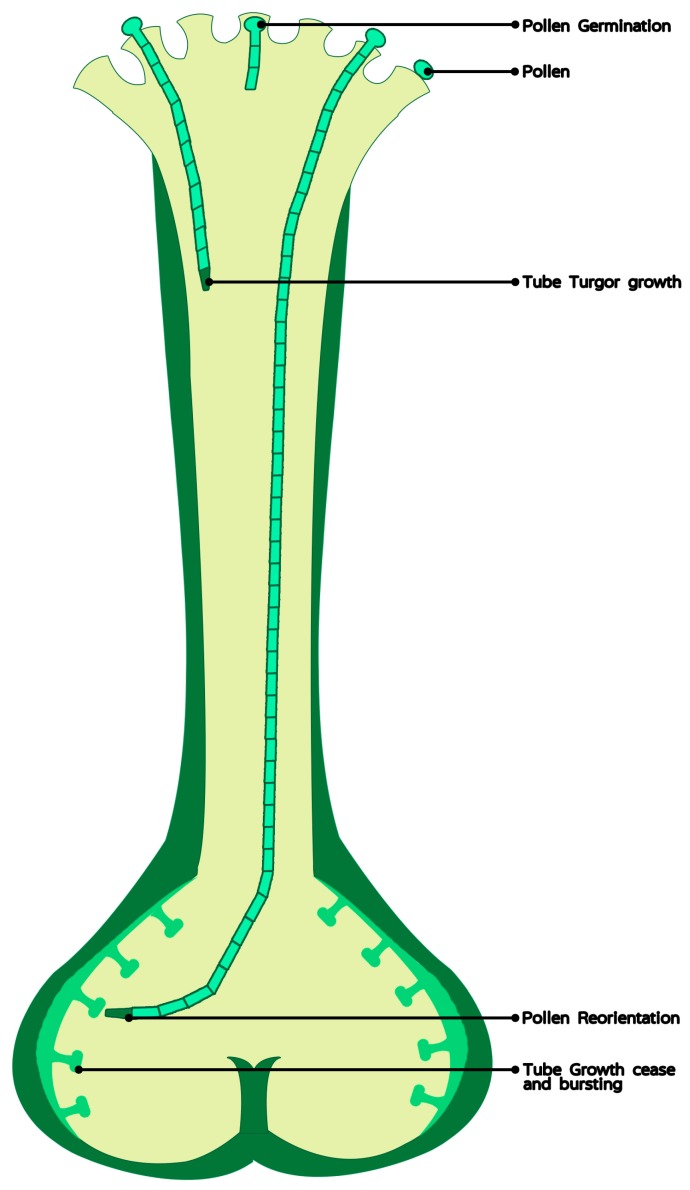
The role of calcium in pollen germination and tube elongation in vivo.

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
