# Peer review of "Calcium: A Critical Factor in Pollen Germination and Tube Elongation"

_ijms, 2019, doi:10.3390/ijms20020420_

Round 1

Reviewer 1 Report

The manuscript falls within the general scope of the journal. However, the paper can be accepted after major revisions. The main problem with manuscript is that some statements are not supported by references.

ABSTRACT

Lines 17-19, Please, specify better. There are a lot of papers concerning function of Ca2+ in plants biology.

INTRODUCTION

Lines 42-43, Please add references for statement: “The results of many studies indicate that Ca2+ dynamics in pollen and the gynoecium are important for these mechanisms to operate.”

Lines 47-49, Please add references: “Although there are continually 48 reports about control mechanisms of pollen tube growth, the most results are from in 49 vitro assays.”

CALCIUM DISTRIBUTION IN THE POLLEN GERMINATION

Lines 102-104, Please add references: “The results of many early studies indicated that Ca2+ is required for the processes of pollen germination and pollen tube growth.”

CALCIUM AND TURGOR FORMATION DURING POLLEN TUBE GROWTH

Lines 165-168, Please add references.

Lines 169-173, Please add references.

Calcium distribution in the stigma

Lines 208-215, Please add references.

CALCIUM DISTRIBUTION IN THE STYLE

Lines 252-253, Please specify better, which flowering plant.

Lines 258-265, Please add references.

CALCIUM DYNAMIC IN SELF-INCOMPATIBILITY RESPONSE

Lines 275-279, Please add references.

This review should be supported with diagrams showing the role of calcium in pollen germination and tube elongation.

Author Response

Reviewer 1

Comments and Suggestions for Authors

The manuscript falls within the general scope of the journal. However, the paper can be accepted after major revisions. The main problem with manuscript is that some statements are not supported by references.

ABSTRACT

Lines 17-19, Please, specify better. There are a lot of papers concerning function of Ca2+ in plants biology.

Answer: Generallyin abstract, no reference is cited.

INTRODUCTION

Lines 42-43, Please add references for statement: “The results of many studies indicate that Ca2+ dynamics in pollen and the gynoecium are important for these mechanisms to operate.”

Answer: have added some words.

Lines 47-49, Please add references: “Although there are continually 48 reports about control mechanisms of pollen tube growth, the most results are from in 49 vitro assays.”

Answer: have added 4 references.

CALCIUM DISTRIBUTION IN THE POLLEN GERMINATION

Lines 102-104, Please add references: “The results of many early studies indicated that Ca2+ is required for the processes of pollen germination and pollen tube growth.”

Answer: have addede 2 references.

CALCIUM AND TURGOR FORMATION DURING POLLEN TUBE GROWTH

Lines 165-168, Please add references.

Answer: as far as we known, no reports about the vacuole formation in pollen tube. We first put out this question. So, no references.

Lines 169-173, Please add references.

Answer: have added a reference.

Calcium distribution in the stigma

Lines 208-215, Please add references.

Answer: It is a very common natural phenomenon, not need references.

CALCIUM DISTRIBUTION IN THE STYLE

Lines 252-253, Please specify better, which flowering plant.

Answer: former sentence has shown “A. majus”.

Lines 258-265, Please add references.

Answer: Have changed.

CALCIUM DYNAMIC IN SELF-INCOMPATIBILITY RESPONSE

Lines 275-279, Please add references.

Answer: the reference is in Lines 281.

 This review should be supported with diagrams showing the role of calcium in pollen germination and tube elongation.

Answer: yes, we have added a diagram. 

Reviewer 2 Report

The manuscript by Zheng et al. summarized the role of Ca2+ and Ca2+ signal in pollen germination and pollen tube elongation during double-fertilization in higher plants. Plant reproduction is a very important topic in plant biology and Ca2+ signaling serves as one of the most critical pathways in the process. In general the manuscript includes the versatile roles of Ca2+ in many aspects during the process. However, the authors mainly reviewed the physiological phenomenon and cellular response. They did not sufficiently discuss the role of Ca2+ in reproduction at the molecular level, which I view it as a very important aspect. There is a good body of current research papers reporting the coding and decoding mechanisms in Ca2+-regulated pollen germination and tube elongation, which are very exciting and significant findings in the field but the authors did not include most of them at all. For instance, I list some of them below:

Bock KW, Honys D, Ward JM, Padmanaban S, Nawrocki EP, Hirschi KD, Twell D, Sze H (2006) Integrating membrane transport with male gametophyte development and function through transcriptomics. Plant Physiol 140: 1151-1168

Frietsch S, Wang YF, Sladek C, Poulsen LR, Romanowsky SM, Schroeder JI, Harper JF (2007) A cyclic nucleotide-gated channel is essential for polarized tip growth of pollen. Proc Natl Acad Sci U S A 104: 14531-14536

Frietsch S, Wang YF, Sladek C, Poulsen LR, Romanowsky SM, Schroeder JI, Harper JF (2007) A cyclic nucleotide-gated channel is essential for polarized tip growth of pollen. Proceedings Of the National Academy Of Sciences Of the United States Of America 104: 14531-14536

Gao QF, Gu LL, Wang HQ, Fei CF, Fang X, Hussain J, Sun SJ, Dong JY, Liu H, Wang YF (2016) Cyclic nucleotide-gated channel 18 is an essential Ca2+ channel in pollen tube tips for pollen tube guidance to ovules in Arabidopsis. Proc Natl Acad Sci U S A 113: 3096-3101

Gutermuth T, Lassig R, Portes MT, Maierhofer T, Romeis T, Borst JW, Hedrich R, Feijo JA, Konrad KR (2013) Pollen tube growth regulation by free anions depends on the interaction between the anion channel SLAH3 and calcium-dependent protein kinases CPK2 and CPK20. Plant Cell 25: 4525-4543

Mahs A, Steinhorst L, Han JP, Shen LK, Wang Y, Kudla J (2013) The calcineurin B-like Ca2+ sensors CBL1 and CBL9 function in pollen germination and pollen tube growth in Arabidopsis. Mol Plant 6: 1149-1162

Michard E, Lima PT, Borges F, Silva AC, Portes MT, Carvalho JE, Gilliham M, Liu LH, Obermeyer G, Feijo JA (2011) Glutamate Receptor-Like Genes Form Ca2+ Channels in Pollen Tubes and Are Regulated by Pistil D-Serine. Science 332: 434-437

Myers C, Romanowsky SM, Barron YD, Garg S, Azuse CL, Curran A, Davis RM, Hatton J, Harmon AC, Harper JF (2009) Calcium-dependent protein kinases regulate polarized tip growth in pollen tubes. Plant J 59: 528-539

Steinhorst L, Mahs A, Ischebeck T, Zhang C, Zhang X, Arendt S, Schultke S, Heilmann I, Kudla J (2015) Vacuolar CBL-CIPK12 Ca(2+)-sensor-kinase complexes are required for polarized pollen tube growth. Curr Biol 25: 1475-1482

Wudick MM, Portes MT, Michard E, Rosas-Santiago P, Lizzio MA, Nunes CO, Campos C, Damineli DSC, Carvalho JC, Lima PT, Pantoja O, Feijo JA (2018) CORNICHON sorting and regulation of GLR channels underlie pollen tube Ca2+ homeostasis. Science 360: 533-536

Xu Y, Yang J, Wang Y, Wang J, Yu Y, Long Y, Wang Y, Zhang H, Ren Y, Chen J, Wang Y, Zhang X, Guo X, Wu F, Zhu S, Lin Q, Jiang L, Wu C, Wang H, Wan J (2017) OsCNGC13 promotes seed-setting rate by facilitating pollen tube growth in stylar tissues. PLoS Genet 13: e1006906

Zhao LN, Shen LK, Zhang WZ, Zhang W, Wang Y, Wu WH (2013) Ca2+-dependent protein kinase11 and 24 modulate the activity of the inward rectifying K+ channels in Arabidopsis pollen tubes. Plant Cell 25: 649-661

Zhou L, Lan W, Chen B, Fang W, Luan S (2015) A calcium sensor-regulated protein kinase, CALCINEURIN B-LIKE PROTEIN-INTERACTING PROTEIN KINASE19, is required for pollen tube growth and polarity. Plant Physiol 167: 1351-1360

Zhou L, Lan W, Jiang Y, Fang W, Luan S (2014) A calcium-dependent protein kinase interacts with and activates a calcium channel to regulate pollen tube growth. Mol Plant 7: 369-376

All these pieces of work (and many others), together, not only uncover the coding of Ca2+ signal by identification of various Ca2+ channels, but also entail some decoding mechanism of Ca2+ signal during fertilization, which connect upstream signal(s) to downstream targets and reactions in a mechanistic manner. In my personal opinion, the authors have to intensively revise the manuscript by reviewing the above-mentioned work as well as other papers. Without the understanding of the topic at the molecular level, the significance of the review will be very limited and thereby may not gain much attention in the field.   

Author Response

Reviewer 2

Comments and Suggestions for Authors

The manuscript by Zheng et al. summarized the role of Ca2+ and Ca2+ signal in pollen germination and pollen tube elongation during double-fertilization in higher plants. Plant reproduction is a very important topic in plant biology and Ca2+ signaling serves as one of the most critical pathways in the process. In general the manuscript includes the versatile roles of Ca2+ in many aspects during the process. However, the authors mainly reviewed the physiological phenomenon and cellular response. They did not sufficiently discuss the role of Ca2+ in reproduction at the molecular level, which I view it as a very important aspect. There is a good body of current research papers reporting the coding and decoding mechanisms in Ca2+-regulated pollen germination and tube elongation, which are very exciting and significant findings in the field but the authors did not include most of them at all. For instance, I list some of them below:

Bock KW, Honys D, Ward JM, Padmanaban S, Nawrocki EP, Hirschi KD, Twell D, Sze H(2006) Integrating membrane transport with male gametophyte development and function through transcriptomics. Plant Physiol 140: 1151-1168

Frietsch S, Wang YF, Sladek C, Poulsen LR, Romanowsky SM, Schroeder JI, Harper JF(2007) A cyclic nucleotide-gated channel is essential for polarized tip growth of pollen. Proc Natl Acad Sci U S A 104: 14531-14536

Frietsch S, Wang YF, Sladek C, Poulsen LR, Romanowsky SM, Schroeder JI, Harper JF(2007) A cyclic nucleotide-gated channel is essential for polarized tip growth of pollen. Proceedings of the National Academy of Sciences of theUnited States of America104: 14531-14536

Gao QF, Gu LL, Wang HQ, Fei CF, Fang X, Hussain J, Sun SJ, Dong JY, Liu H, Wang YF(2016) Cyclic nucleotide-gated channel 18 is an essential Ca2+ channel in pollen tube tips for pollen tube guidance to ovules in Arabidopsis. Proc Natl Acad Sci U S A 113: 3096-3101

Gutermuth T, Lassig R, Portes MT, Maierhofer T, Romeis T, Borst JW, Hedrich R, Feijo JA, Konrad KR (2013) Pollen tube growth regulation by free anions depends on the interaction between the anion channel SLAH3 and calcium-dependent protein kinases CPK2 and CPK20. Plant Cell 25:4525-4543

Mahs A, Steinhorst L, Han JP, Shen LK, Wang Y, Kudla J (2013) The calcineurin B-like Ca2+ sensors CBL1 and CBL9 function in pollen germination and pollen tube growth in Arabidopsis. Mol Plant 6: 1149-1162

Michard E, Lima PT, Borges F, Silva AC, Portes MT, Carvalho JE, Gilliham M, Liu LH, Obermeyer G, Feijo JA (2011) Glutamate Receptor-Like Genes Form Ca2+ Channels in Pollen Tubes and Are Regulated by Pistil D-Serine. Science 332: 434-437

Myers C, Romanowsky SM, Barron YD, Garg S, Azuse CL, Curran A, Davis RM, Hatton J, Harmon AC, Harper JF (2009) Calcium-dependent protein kinases regulate polarized tip growth in pollen tubes. Plant J 59: 528-539

Steinhorst L, Mahs A, Ischebeck T, Zhang C, Zhang X, Arendt S, Schultke S, Heilmann I, Kudla J (2015) Vacuolar CBL-CIPK12 Ca(2+)-sensor-kinase complexes are required for polarized pollen tube growth. Curr Biol 25: 1475-1482

Wudick MM, Portes MT, Michard E, Rosas-Santiago P, Lizzio MA, Nunes CO, Campos C, Damineli DSC, Carvalho JC, Lima PT, Pantoja O, Feijo JA (2018) CORNICHON sorting and regulation of GLR channels underlie pollen tube Ca2+ homeostasis. Science 360: 533-536

Xu Y, Yang J, Wang Y, Wang J, Yu Y, Long Y, Wang Y, Zhang H, Ren Y, Chen J, Wang Y, Zhang X, Guo X, Wu F, Zhu S, Lin Q, Jiang L, Wu C, Wang H, Wan J (2017) OsCNGC13 promotes seed-setting rate by facilitating pollen tube growth in stylar tissues. PLoS Genet 13:e1006906

Zhao LN, Shen LK, Zhang WZ, Zhang W, Wang Y, Wu WH (2013) Ca2+-dependent protein kinase11 and 24 modulate the activity of the inward rectifying K+ channels in Arabidopsis pollen tubes. Plant Cell 25: 649-661

Zhou L, Lan W, Chen B, Fang W, Luan S (2015) A calcium sensor-regulated protein kinase, CALCINEURIN B-LIKE PROTEIN-INTERACTING PROTEIN KINASE19, is required for pollen tube growth and polarity. Plant Physiol 167: 1351-1360

Zhou L, Lan W, Jiang Y, Fang W, Luan S (2014) A calcium-dependent protein kinase interacts with and activates a calcium channel to regulate pollen tube growth. Mol Plant 7: 369-376

All these pieces of work (and many others), together, not only uncover the coding of Ca2+ signal by identification of various Ca2+ channels, but also entail some decoding mechanism of Ca2+ signal during fertilization, which connect upstream signal(s) to downstream targets and reactions in a mechanistic manner. In my personal opinion, the authors have to intensively revise the manuscript by reviewing the above-mentioned work as well as other papers. Without the understanding of the topic at the molecular level, the significance of the review will be very limited and thereby may not gain much attention in the field.   

Answer: there are many reports about pollen tube growth each year. Our review focus on two points: calcium distribution characteristic and interaction of male and female cells and tissues in vivo, to speculate calcium physiological function during pollen tube growth. We believe that Ca2+ signal exists during fertilization (during tube reorientation), but we discuss distribution characters of calcium in pollen tube and female cells in this review, which is characteristic of our review. We mainly focus on the pollen tube growth in vivo (see Conclusion and prospects), which be ignored by most reports. In addition, we discuss the calcium role in whole process of pollen tube growth in vivo. On the contrary, related molecular works focus on one point of pollen tube growth, which is not our purport. Actually, these molecular works mainly focus on genes and proteins, not on calcium. Adding molecular references will let our topic off tracking, and make our review too long.  

Round 2

Reviewer 1 Report

Manuscript was significantly improved.

Reviewer 2 Report

Personally I still think this review is mostly descriptive on phenomenon. There is a Ca2+ signal and physiological response in every pollen-related physiological process. However, whether the correlation is really tight and falls into the cause-effect relationship is not that clear in most of the studies. The genes and proteins which I mentioned are actually most critical in understanding the mechanisms at the molecular lever. Since the other reviewer had no objection, I would accept the authors' argument and let it go for publication.